# Beyond Individual Acute Phase Protein Assessments: Introducing the Acute Phase Index (API) as a Prognostic Indicator in Dogs with Malignant Neoplasia

**DOI:** 10.3390/vetsci12060533

**Published:** 2025-06-01

**Authors:** Martina Baldin, Maria Elena Gelain, Giacomo Marolato, Silvia Bedin, Michele Berlanda, Manuela Zanetti, Filippo Torrigiani, Alessia Giordano, Pierangelo Moretti, Donatella Scavone, Federico Bonsembiante

**Affiliations:** 1Department of Comparative Biomedicine and Food Science, University of Padua, 35020 Legnaro, Italy; martina.baldin.2@phd.unipd.it (M.B.); filippotorrigiani6@gmail.com (F.T.); 2Istituto Zooprofilattico Sperimentale del Lazio e della Toscana, 00100 Rome, Italy; marolatogiacomo@gmail.com; 3Department of Animal Medicine, Production and Health, University of Padua, 35020 Legnaro, Italy; silvia.bedin@unipd.it (S.B.); michele.berlanda@unipd.it (M.B.); federico.bonsembiante@unipd.it (F.B.); 4Antech Diagnostics Italy S.r.l., 20100 Milan, Italy; manuela.zanetti@antechdx.com; 5Department of Veterinary Medicine and Animal Sciences, University of Milan, 26900 Lodi, Italy; alessia.giordano@unimi.it (A.G.); donatella.scavone@unimi.it (D.S.); 6Department of Veterinary Sciences, University of Turin, 10095 Grugliasco, Italy; pierangelo.moretti@unito.it

**Keywords:** acute phase proteins, acute phase index, dog, neoplasia, clinical pathology, oncology

## Abstract

Cancer can alter the levels of acute phase proteins (APPs), which are involved in the innate immune response characterized by high sensitivity and low specificity. Combining APPs into an Acute Phase Index (API) enhanced their utility in monitoring human disease progression. In veterinary medicine, APIs have only been applied to livestock and dogs with Leishmaniasis. This study aimed to calculate an API to assess its value in cancer-bearing dogs. Serum samples were collected from 55 dogs, and multiple samples were available in 32 dogs. Patients were classified by neoplastic category and survival times (> or <30 and 90 days). The API included two positive and two negative APPs. Lower Paraoxonase-1 (PON-1) levels were found in round-cell tumors, possibly reflecting higher oxidative stress. Moreover, API increased and PON-1 activity decreased in the last sample in dogs that died before the end of the study. Dogs with elevated APIs and C-reactive protein had shorter survival times. An API greater than 0.049 at first sampling was associated with a 3.7-fold higher risk of death. These findings highlight the prognostic potential of APIs in canine cancer.

## 1. Introduction

Acute phase proteins (APPs) are plasma proteins with varying concentrations or activities during inflammatory diseases [1]. Positive APPs (PAPPs) include proteins and other molecules with concentrations that increase in response to inflammatory cytokines, such as IL-1, IL-6, and TNF-α, and are primarily synthesized by hepatocytes [1,2]. Negative APPs (NAPPs) are proteins with decreased serum or plasma concentration or activity due to an inflammatory condition [1].

PAPPs are further classified as major, moderate, or minor based on the magnitude and duration of their response. Major PAPPs increase 100–1000 times and normalize within 48 h post-resolution; moderate PAPPs rise 5–10 times over 2–3 days, decreasing more gradually; and minor PAPPs show a gradual 5–100% increase from baseline [3].

APPs are part of the innate immune response; their high sensitivity allows them to be used as early markers of inflammatory processes, but their low specificity [4] prevents them from localizing the ongoing pathological process [3].

The main PAPPs are C-reactive protein (CRP), serum amyloid A (SAA), α-1 acid glycoprotein (AGP), haptoglobin (Hp), ceruloplasmin (Cp), and fibrinogen [4]. In the canine species, the most commonly measured PAPPs are CRP as a major APP and Hp as a moderate APP. CRP is involved in the opsonization and destruction of bacterial microorganisms, while Hp binds free hemoglobin to prevent the loss of iron and reduce its availability to microorganisms [1].

The main NAPP is albumin, as the liver prioritizes the production of positive acute phase proteins over albumin during the acute phase response [2,5]. Another NAPP that has recently gained interest is Paraoxonase-1 (PON-1), which has anti-inflammatory properties and protects low- and high-density lipoproteins from lipid peroxidation [6,7,8].

Neoplastic conditions often lead to increased PAPPs for various reasons, including production by neoplastic cells, tumor-induced inflammation, or release from surrounding tissues damaged by the neoplasm, particularly in malignant and rapidly expanding tumors [1,9]. This response may also arise from metastases, ulcerative processes, or secondary infections linked to immunosuppression [10]. Several studies have evaluated changes in APPs due to neoplastic conditions, their severity, and prognostic usefulness [9,11,12,13,14,15,16,17,18].

The degree of inflammation in neoplastic conditions influences the acute phase response, reflecting non-specific inflammation due to local necrosis or tissue damage [18]. For instance, a study on dogs with mammary tumors showed significantly increased CRP and SAA during metastatic dissemination, although CRP levels did not correlate with prognostic factors [17]. Additionally, high CRP levels were observed in canine mast cell tumors (MCTs), likely due to the chemotactic role of APPs in mast cell activation. The same study found elevated CRP, Hp, and AGP concentrations in dogs with sarcomas [11]. Increased CRP and Hp levels have also been found in dogs with high-grade multicentric lymphoma, acute lymphocytic leukemia, and multiple myeloma [16,19,20]. In dogs with circum-anal gland tumors, both benign and malignant, increased SAA, CRP, and HP levels have been observed, with higher elevations within carcinomas [15]. Additionally, in the same patients, decreased APPs have been noted after anti-tumor therapy, coinciding with reduced clinical signs, tumor sizes, and the cessation of tumor-related bleeding [15].

Although some studies have assessed multiple APPs, most of the relevant literature has focused on individual markers. These studies have shown limited success thus far in defining specific clinical or prognostic applications [21]. An index that includes multiple APPs (i.e., an Acute Phase Index, API) has been proposed to overcome this problem, which includes both positive and negative APPs [22], according to the following mathematical formula: API = (positive APP “fast” * positive APP “slow”)/(negative APP “fast” * negative APP “slow”) [21]. For the positive APPs, “fast” and “slow” are synonyms for “major” and “moderate”.

In human medicine, various indices combining inflammatory and nutritional markers have been developed, such as the Prognostic Inflammatory and Nutritional Index (PINI) and the Prognostic Nutritional Index (PNI). These are primarily used for prognostic assessment in specific clinical settings, including with geriatric patients, critically ill individuals, and those with neoplastic diseases, such as multiple myeloma or renal carcinoma [23,24,25].

In veterinary medicine, APIs have only been evaluated in bovine species [26], swine [26,27], and dogs affected by Leishmaniasis [28]. In cattle, an index combining SAA, Hp, albumin, and α2-macroglobulin was proposed; this API represented a more robust indicator of animal health status, particularly in herd-level applications [26]. In pigs, various APP combinations were tested for their ability to detect infectious diseases. The study found that APP combinations provided higher sensitivity than single markers, with the best-performing three-protein combinations including CRP, apolipoprotein A1 (apoA1), and pig major acute phase protein (pigMAP) or Hp. Two-protein combinations, such as CRP or apoA1 with pigMAP, also showed stronger performance than that when evaluating a single APP, supporting the value of multivariate APP-based approaches in swine health surveillance [27]. In canine medicine, two different APIs were calculated in a group of dogs with Leishmaniasis to assess which would be the most reliable; in particular, an index that combines a positive and a negative APP and another index that includes two positive and one negative APP were used [26]. The results of these evaluations showed a slightly greater reliability of the second index ((CRP * Cp)/Alb) in the onset of a suspected diagnosis and therapy monitoring [28].

Despite the encouraging results across these species, API-based tools in veterinary medicine remain limited and would benefit from further validation in different clinical and production contexts.

These studies demonstrate how employing an index of both positive and negative acute phase proteins enhances their capacity for diagnosis, monitoring, and prognostic evaluation.

In veterinary medicine, no studies evaluating API in cancer-bearing patients are available to date. Thus, the aim of this work was to calculate an API in dogs with neoplastic diseases and subsequently assess its prognostic significance, in addition to the individual APP analysis [29,30].

## 2. Materials and Methods

This is a prospective study. The patients selected for this study were dogs presented to the Oncology Unit of the Veterinary Teaching Hospital (VTH) between 2019 and 2021, diagnosed with a malignant neoplastic disease through cytology, histology, flow cytometry, or a combination of these methods, performed either in the internal laboratory or an external one.

This study was structured in two phases: (a) a cross-sectional phase, which included patients from whom only one sample was available; (b) a longitudinal phase, involving patients with multiple samples, in which changes in analyte concentrations between the first and the last sampling were assessed. In the cross-sectional phase, differences among tumor types and survival times were evaluated, diagnostic accuracy for survival was calculated, and survival analysis was performed.

Ethical approval was not required, as the serum samples were residuals from routine diagnostics, and procedures complied with Legislative Decree 26/2014 (Directive 2010/63/EU). Owners provided written informed consent for data processing.

Samples were obtained via jugular venipuncture, allowed to clot for 15 min, and centrifuged at 3000× *g* for 10 min. The serum was collected, frozen, and stored at −20 °C until analysis. Visibly lipemic, hemolytic, or jaundiced samples were excluded.

Patients were categorized into four groups based on the cytomorphological classification of neoplasms: epithelial, spindle-cell mesenchymal, round-cell mesenchymal, and other neoplasms (including melanocytic tumors and those for which classification was not possible due to poor differentiation).

Additionally, for survival assessment, animals were divided based on survival time, with cut-offs set at 30 and 90 days of survival from the diagnosis.

CRP, albumin, and haptoglobin levels were measured at the clinical pathology laboratory of the VTH of the University of Padua, Italy, using an automatic analyzer for liquid clinical biochemistry (BT 3500, Biotecnica Instruments S.p.A., Rome, Italy). The CRP level was determined using a turbidimetric assay “Biotecnica”, utilizing a goat polyclonal antibody against human CRP as a reagent, with previous validation performed in the same laboratory [31]. The albumin concentration was determined using a colorimetric method, employing a commercially available bromocresol green reagent. Haptoglobin concentration was determined using a turbidimetric method with a commercially available kit (Gesan S.r.l., Campobello di Mazara, Italy). This assay was previously validated in the same laboratory [29].

PON-1 measurement was performed at the clinical pathology laboratory of the VTH of the University of Milan. An automatic analyzer for liquid clinical biochemistry (Cobas Mira, Roche Diagnostic, Basel, Switzerland) was utilized, employing a spectrophotometric enzymatic method previously validated in dogs in the same laboratory [8].

PON-1 activity values were transformed into conversion factors for API calculation, as PON-1 is measured by enzymatic activity, unlike other APPs measured in serum concentration. The conversion was performed dividing PON-1 data into quartiles: values above 181 U/L (threshold between second and third quartile) were assigned a conversion factor of 4; values between 155.7 U/L (threshold between first and second quartile) and 181 U/L were assigned a conversion factor of 3; values between 106.6 U/L (lower limit of the reference interval) and 155.7 U/L were assigned a conversion factor of 2; and values below 106.6 U/L were assigned a conversion factor of 1.5. Greater consideration was given to the left side of the PON-1 value distribution curve when assigning conversion factors to achieve greater stratification in the lower values, given that PON-1 is an NAPP. Conversely, we used 1.5 as the conversion factor instead of 1 to prevent the API calculation from being identical to that without PON-1 in patients with lower PON-1 levels, as the conversion factor is in the denominator of the formula. Table 1 provides the cut-offs and corresponding conversion factors.

The API was calculated as follows [21]:$$\frac{CRP\;\left(\frac{g}{L}\right)\times Hp\;\left(\frac{g}{L}\right)}{Alb\;\left(\frac{g}{L}\right)\times PON-1\;\left(conversion\;factor\right)}\times 1000$$

Additionally, an API was calculated without considering PON-1, as follows:$$\frac{CRP\;\left(\frac{g}{L}\right)\times Hp\;\left(\frac{g}{L}\right)}{Alb\;\left(\frac{g}{L}\right)}\times 1000$$

Differences in APPs and APIs depending on neoplasia type were assessed using the Kruskal–Wallis test, evaluating the analytes on the first sample collected from each animal. The Mann–Whitney test was used to evaluate differences in APPs and APIs in relation to survival time. Comparisons were made between dogs that survived less than and more than 30 days, as well as between dogs that survived less than and more than 90 days. Receiver operating characteristic (ROC) curves were developed for each analyte to evaluate their diagnostic accuracy, using the survival status of the animals at the end of the study as the discriminating factor. Subsequently, a Kaplan–Meier curve was developed for survival analysis using the API value that demonstrated the highest diagnostic specificity and sensitivity determined by the ROC curve.

The Wilcoxon test assessed differences in APPs and APIs between the first and last sampling in living or deceased animals subjected to multiple samplings. Differences were considered significant at *p* < 0.05.

Statistical analysis was performed using the MedCalc Statistical Software (version 19.2.6, 2020, Mariakerke, Belgium), while figures were generated with the R Statistical Software (version 4.4.2, GUI 1.81, Big Sur ARM build, 2023).

## 3. Results

### 3.1. Samples and Study Population

During the study period, fifty-five dogs were enrolled in the cross-sectional phase. In total, 22 underwent multiple samplings throughout their clinical course and were subsequently included in the longitudinal phase, resulting in 117 samples. Twenty-one dogs died, with ten having undergone multiple samplings.

The dogs in the study ranged in age from 1 to 18 years, with a mean and median of 9.1 and 9 years, respectively. There were seven intact females, twenty-six spayed females, twenty-one intact males, and four neutered males.

The definitive diagnosis of malignant neoplastic disease was established through microscopic evaluation of the lesions, utilizing cytological and/or histological analyses, and, in some cases, supplemented with flow cytometry, as summarized in Table 2.

There were two exceptions to this approach: (1) the intracranial tumor was the only case assessed solely with magnetic resonance imaging (MRI), because of the inability to sampling due to its pros-encephalic localization; and (2) one case of suspected chronic lymphocytic leukemia or stage V lymphoma was included in the study based solely on complete blood count and peripheral blood film evaluation, due to the owner’s unwillingness to pursue further investigations.

Out of the fifty-five patients, ten had epithelial neoplasms (five squamous cell carcinoma, four transitional cell carcinoma, and one prostatic carcinoma), with four undergoing multiple samplings, resulting in a total of seventeen samples. Sixteen patients had spindle cell mesenchymal neoplasms (seven soft tissue sarcomas, one fibrosarcoma, three hemangiosarcomas, one osteosarcoma, one undifferentiated sarcoma, one intracranial tumor, one splenic stromal sarcoma, and one gastric leiomyosarcoma or gastrointestinal stromal tumor), with five undergoing multiple samplings, for a total of twenty-four samples. The intracranial tumor was classified among spindle-cell mesenchymal neoplasms as the most likely diagnosis based on the diagnostic imaging report provided by another facility. Immunohistochemistry was recommended for a definitive diagnosis of gastric leiomyosarcoma or gastrointestinal stromal tumor (GIST). However, the owner declined, leaving the two conditions as differential diagnoses. Twenty-seven patients had round-cell neoplasms (eleven lymphoid neoplasia, eleven mast cell tumors, one concurrent lymphoma and mast cell tumor, two plasma cell tumors, one multiple myeloma, and one histiocytic sarcoma), with eleven undergoing multiple samplings, for a total of seventy-three samples. One patient had an oral melanoma and one a poorly differentiated malignant neoplasm. Thus, they were categorized as “other neoplasms”. However, based on the low number of dogs, this group was not included in the statistical analysis regarding neoplastic type. These numbers are shown in Table 3.

### 3.2. APPs and API

The results obtained from APP measurement and the calculated APIs across all samples for each neoplastic category are presented in Table 4.

#### 3.2.1. Cross-Sectional Study

##### Neoplasia Type

PON-1 was significantly lower in round-cell mesenchymal neoplasms, mostly represented by lymphoma and mast cell tumor, compared to epithelial and spindle-cell mesenchymal neoplasms (*p* = 0.049) (Table 5 and Figure 1). Additionally, despite no significant overlap with the other groups, API reached higher values in epithelial neoplasms, mostly represented by squamous and transitional cell carcinomas, compared to round- and spindle-cell mesenchymal neoplasms (Table 5 and Figure 1).

##### Survival Time

API and API without PON-1 were significantly higher in the group of animals with shorter survival times (i.e., less than 30 or 90 days), compared to the group with longer survival times (for 30 and 90 days, *p* = 0.001 and *p* = 0.013 for API and *p* < 0.001 and *p* = 0.005 for API without PON-1, respectively) (Table 6, Figure 2). CRP and Hp were significantly higher in the group of dogs with a survival time of less than 30 days compared to those with a survival time of more than 30 days (*p* < 0.001 and *p* = 0.016, respectively); moreover, CRP was significantly higher in dogs with a survival time of less than 90 days compared to those with a survival time of more than 90 days (*p* = 0.011).

##### ROC Curves and Survival Analysis

CRP, API, and API without PON-1 demonstrated the highest diagnostic accuracy for survival, with AUC values exceeding 0.7 (Figure 3). Detailed results are presented in Table 7.

Kaplan–Meier survival analysis was performed using the APIs with and without PON-1, applying cut-off values that provided the best sensitivity and specificity based on the ROC curve. Figure 4 illustrates the relationship between survival probability and observation time, stratified by whether the API at the first sampling was above or below 0.049, and whether the API without PON-1 at the first sampling was above or below 0.202 (Table 7).

There was a significant difference (*p* = 0.0110 and *p* = 0.0032) between the curves in both cases. An API > 0.049 at first sampling showed a 3.7 times higher probability of dying compared to animals with an API at first sampling < 0.049. By the study endpoint, 81.82% of animals with an API < 0.049 were still alive (Table 8). Regarding API without PON-1, values > 0.202 at first sampling revealed a 4.4 times higher probability of dying compared to animals with API without PON-1 at first sampling < 0.202. By the study endpoint, 83.33% of animals with an API < 0.202 were still alive (Table 8).

#### 3.2.2. Longitudinal Study

The difference between the first and last samples in animals subjected to multiple sampling was also evaluated, considering their outcome at the end of the study (survivors, *n* = 12; non-survivors, *n* = 10). In non-surviving animals, significantly increased API and decreased PON-1 were observed between the first and last sampling (*p* = 0.019 and 0.013, respectively) (Figure 5).

## 4. Discussion

APPs are important indicators of animal health status, as they represent the non-specific response of the organism to many pathological processes [32]. However, the low specificity of APPs makes them less useful when considered individually, especially in complex conditions, such as neoplasms associated with other concurrent diseases. In addition, cancer-bearing patients could have an altered immune response and are, therefore, more susceptible to secondary infections, possibly affecting the acute phase response [32].

In the present study, the individual analytes—CRP, albumin, Hp, and PON-1—and two different APIs were evaluated: the first included all the aforementioned APPs, while the second excluded PON-1. We decided to calculate a second API without PON-1 because its values had been replaced using conversion factors. As these factors were chosen arbitrarily, we thought it would be useful to calculate one without this analyte.

Epithelial, spindle-cell mesenchymal, and round-cell mesenchymal neoplasms were compared to evaluate the possible influence of tumor type on the acute phase response. PON-1 was significantly lower in round-cell mesenchymal neoplasms, with lymphoma and mast cell tumor being the more common tumors in this category. This result could be explained by a greater level of oxidative stress caused by free radical production, which is reportedly greater in lymphoma [33]. Additionally, although not statistically significant, API was mildly higher in epithelial neoplasms, which were mostly represented by squamous and transitional cell carcinomas in this study. This suggests that the acute response could be greater in these cancers. Indeed, malignant epithelial neoplasms usually have a large inflammatory component and possible ulceration [10,11]. Squamous cell carcinomas are often erythematous and ulcerated, leading to possible secondary bacterial infection of the lesion [34].

The study demonstrated that both versions of the API and CRP were significantly higher in animals with shorter survival times (<30 and <90 days), supporting the hypothesis that the severity of tumor-induced inflammation has prognostic relevance. Neoplasms associated with a more intense inflammatory response appear to have a worse clinical course, resulting in reduced survival. It is also likely that these animals were in a more advanced clinical condition at the time of sampling [10].

Regarding Hp, a significant increase was observed in animals with a survival of <30 days than ≥30 days. However, no significant difference was found when comparing animals with survival < 90 vs. ≥90 days. CRP showing significant differences at both 30- and 90-day cut-offs, while Hp was only at 30 days, may reflect the higher sensitivity of CRP as an acute phase protein. Conversely, Hp only increased in animals with the shortest survival, presumably those with a more severe inflammatory burden.

It is worth noting that sample timing was not standardized but depended on the clinical evaluation schedule. Therefore, some samples may have been collected before a significant increase in Hp, whereas CRP—a more rapidly responding marker—had already reached detectable elevations.

According to the ROC curve analysis, CRP and APIs with and without PON-1 demonstrated better diagnostic accuracy for survival. Survival probability analysis revealed that an API value higher than 0.049 at the first sampling was associated with a 3.7-fold higher probability of death than animals with API values below 0.049. Similarly, an API without PON-1 exceeding 0.202 at the first sampling was linked to a 4.4-fold higher probability of death compared to those with an API without PON-1 values below 0.202. Notably, 81% of dogs with an API < 0.049 and 83% of dogs with an API without PON-1 < 0.202 survived until the end of the study, supporting the potential use of APIs as prognostic factors.

In animals included in the longitudinal study, significant differences in variables between the first and last sampling were only highlighted in non-surviving animals: significantly increased API was noted, revealing a progressive inflammatory component in deceased patients. Several mechanisms underlie the changes in APP concentration or activity observed in cancer. The rise in PAPPs is not only driven by tumor-induced inflammation but also by direct production from neoplastic cells and release from surrounding tissues damaged by the tumor [1]. Decreased NAPPs, such as albumin and PON-1, can be attributed to different mechanisms: albumin synthesis is downregulated as the liver prioritizes acute phase protein production, and its reduction may be further exacerbated by cancer-associated cachexia development. Meanwhile, PON-1 levels decrease in response to oxidative stress, which could be associated with tumor progression [33,35,36,37]. It is possible that non-surviving patients had larger and/or particularly aggressive, rapidly expanding lesions, leading to a sustained and progressively increasing acute-phase response throughout the study.

Moreover, PON-1 significantly decreased between the first and last sampling in non-surviving animals. This decrease is attributable to the nature of PON-1 as an NAPP, which tends to decline in conditions marked by excess reactive oxygen species (ROS), including those produced by neoplastic cells. Free radicals, by modulating gene expression in carcinogenesis [38,39], indirectly induce the release of cytokines which, in turn, attract other inflammatory cells [40] that contribute to excess reactive oxygen species. This creates a vicious circle that contributes to neoplastic progression [40].

The results suggest that the acute phase response plays a very important role in neoplastic pathology progression and that there is a basis for introducing APP evaluation into clinical oncology practice for prognostic purposes, as already achieved in human medicine [23]. Given that different acute phase proteins exhibit distinct behaviors based on tumor type, it would be appropriate to assess multiple APPs to obtain a more comprehensive evaluation of the oncologic patient.

Regarding the limitations of this study, the inclusion of PON-1, quantified by enzyme activity rather than serum concentration, required conversion factors in one of the two indices. These factors ranged from 1.5 to 4, which greatly reduced the measured data variability. The API without PON-1 was assessed to determine whether the applied correction factors resulted in similar trends between the two APIs. In future studies, it would be more appropriate to directly measure enzyme concentrations rather than activity. Another option could be to increase the number of conversion factors utilized.

The decision to include dogs with only a cytologic diagnosis of malignant neoplasm allowed us to obtain a larger sample size but also required broader categories to distinguish between neoplastic types. Thus, categorizing neoplastic types was based on cytomorphologic categories, which include neoplasms with very different biological behaviors, invasion and metastasis mechanisms, and severities that cannot generally be attributed to the macro-category. This categorization aimed to help assess whether different neoplasm types influence the acute phase response. However, the reasons for the observed differences remain unclear and have only been hypothesized. To better understand the role of the acute phase response in various tumors, further studies could analyze specific neoplasms in more detail, including subgroups based on tumor grade and staging.

It would also be interesting to investigate APPs and API in animals grouped according to the clinical protocol they are subjected to; for example, animals undergoing chemotherapy, palliative care, surgery, and no treatment.

Moreover, the sample timing was not standardized, and the overall number of subjects was relatively small and distributed across a wide range of tumor types, which may trigger different acute phase responses and limit the generalizability of the findings.

## 5. Conclusions

Inflammation, as part of the so-called “hallmarks of cancer”, has a fundamental role in malignant neoplasm evolution [40]. Therefore, both the extent of damage due to the neoplastic process and cytokine production by neoplastic cells can affect APP production. This study highlighted the importance of evaluating the acute phase response as a whole rather than individually assessing APPs, as this often does not yield perfectly valid results.

The most significant findings concerning API use in this study emerged particularly concerning survival: overall, animals with shorter survival times exhibited higher API values, suggesting a more pronounced acute phase response. Furthermore, survival analysis revealed that API values at the first sampling significantly impacted survival time. Specifically, an API > 0.049 was associated with a 3.7-fold higher probability of death, while an API without PON-1 > 0.202 corresponded to a 4.4-fold higher probability of death, compared to animals with APIs with and without PON-1 values below these respective thresholds.

These results support the hypothesis that including more analytes in the composition of an index enhances the prognostic power of APPs in evaluating clinical outcomes.

In conclusion, our results can provide a baseline for possible future projects and developments, and it would be very interesting to understand in which specific neoplasms the API could be considered a solid prognostic factor.

## Figures and Tables

**Figure 1 vetsci-12-00533-f001:**
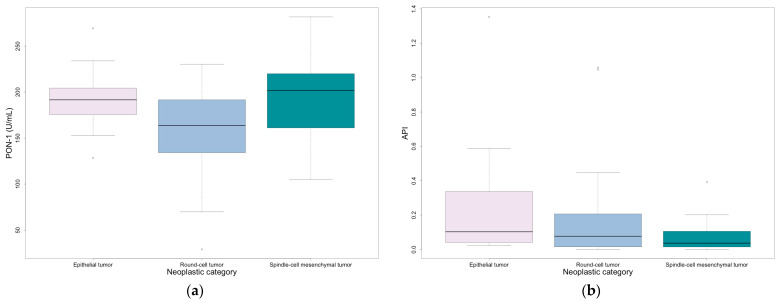
Values at first sampling divided by neoplastic category: (**a**) PON-1; (**b**) API. Outliers are represented by circles.

**Figure 2 vetsci-12-00533-f002:**
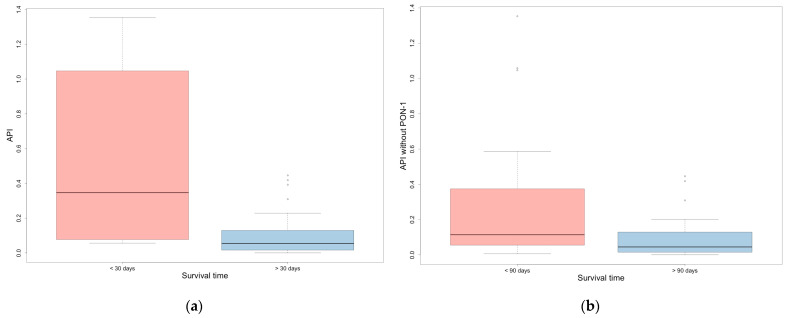
(**a**) API in dogs that survived more or less than 30 days; (**b**) API without PON-1 in dogs that survived more or less than 90 days. Outliers are represented by circles.

**Figure 3 vetsci-12-00533-f003:**
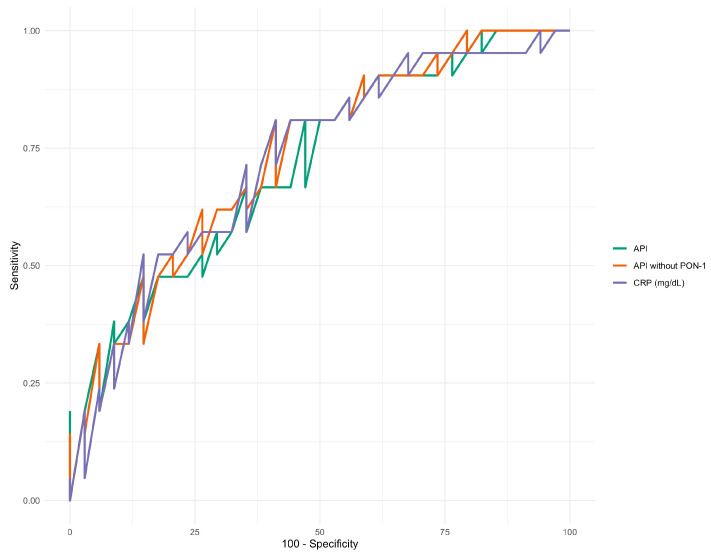
ROC curves for CRP and APIs with and without PON-1.

**Figure 4 vetsci-12-00533-f004:**
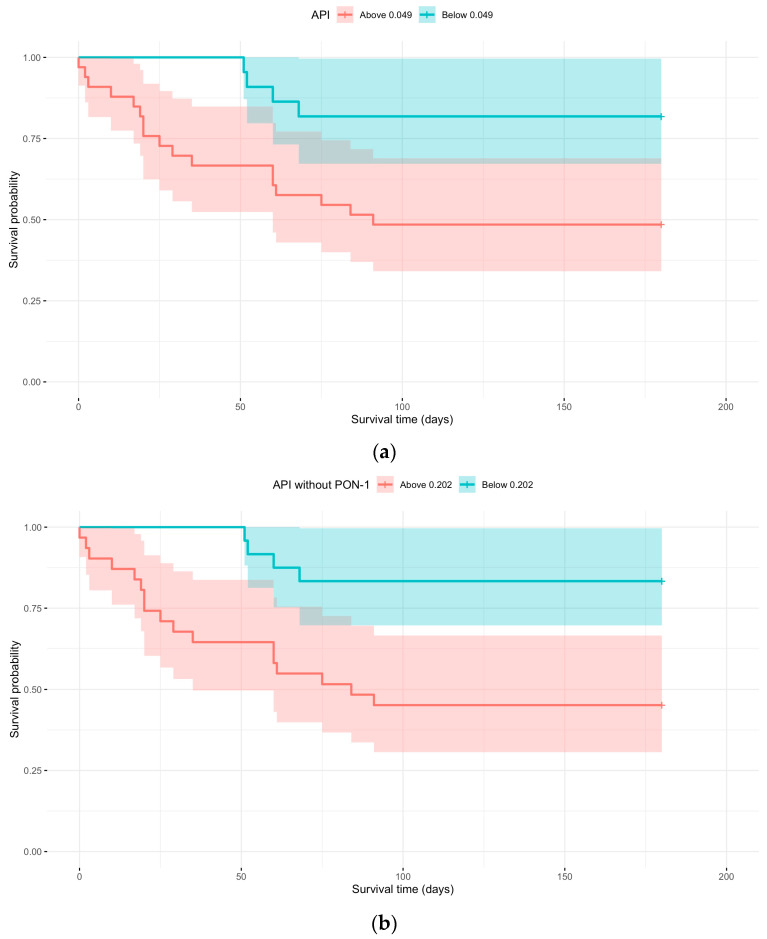
(**a**) Kaplan–Meier curve with API above or below 0.049 at first sampling; (**b**) Kaplan–Meier curve with API without PON-1 above or below 0.202 at first sampling.

**Figure 5 vetsci-12-00533-f005:**
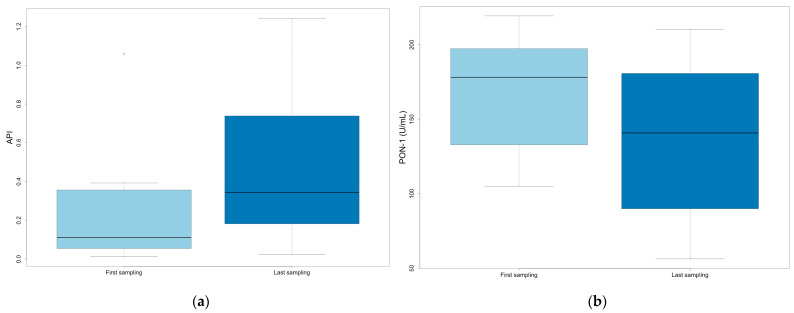
Values at first and last sampling in non-surviving animals: (**a**) API; (**b**) PON-1.

**Table 1 vetsci-12-00533-t001:** PON-1 values and corresponding conversion factors.

PON-1 (U/mL)	Conversion Factor
<106.6	1.5
106.6–155.6	2
155.7–181.0	3
>181.0	4

**Table 2 vetsci-12-00533-t002:** Diagnosis method.

Method	Number of Cases
Cytology	18
Histology	10
Flow cytometry	1
Cytology + histology	12
Cytology + flow cytometry	7
Diagnosis at an external laboratory (cytology or histology)	5
Diagnostic imaging (external facility)	1
CBC and peripheral blood film assessment	1

**Table 3 vetsci-12-00533-t003:** Dogs categorized by neoplastic type, number of diagnoses of animals subjected to multiple samplings, and consequent number of samples for each category.

Type of Neoplasia		Number of Patients	Number of Patients with Multiple Sampling	Number of Samples
Epithelial		10	4	17
	Squamous Cell Carcinoma	5	2	9
	Transitional Cell Carcinoma	4	2	7
	Prostatic Carcinoma	1	0	1
Spindle-cell mesenchymal		16	5	24
	Soft Tissue Sarcoma	7	1	8
	Fibrosarcoma	1	1	2
	Splenic Stromal Sarcoma	1	1	2
	Hemangiosarcoma	3	2	8
	Osteosarcoma	1	0	1
	Undifferentiated sarcoma	1	0	1
	Intra-cranic tumor	1	0	1
	Leiomyosarcoma	1	0	1
Round-cell mesenchymal		27	11	73
	Lymphoma	10	6	31
	Mast Cell Tumor	11	4	31
	Chronic lymphocytic leukemia/stage V lymphoma	1	1	4
	Lymphoma + Mast Cell Tumor	1	0	1
	Multiple Myeloma	1	1	3
	Plasma Cell Tumor	2	0	2
	Histiocytic Sarcoma	1	0	1
Others		2	1	3
	Oral Melanoma	1	1	2
	Poorly Differentiated Malignant Neoplasm	1	0	1

**Table 4 vetsci-12-00533-t004:** Descriptive statistics for acute phase proteins and Acute Phase Indexes divided by neoplastic category, reported as median and range (minimum–maximum).

Type of Neoplasia	E *	S *	R *	O *
N *	17	24	73	3
CRP (mg/dL)	0.774 (0.339–4.17)	0.514 (0.136–1.700)	0.703 (0.060–3.590)	0.732 (0.469–2.520)
Hp (mg/dL)	111.6 (1.0–173.2)	63.0 (1.0–151.2)	130.7 (1.0–173.4)	158.0 (155.0–168.0)
Alb (g/L)	30.28 (19.90–35.64)	28.69 (16.26–29.4)	32.68 (16.63–40.73)	26.19 (20.96–26.58)
PON-1 (U/mL)	200.7 (56.2–269.2)	200.7 (56.2–325.6)	180.6 (29.0–327.1)	180.6 (29.0–327.1)
API	0.076 (0.001–1.353)	0.028 (0.000–0.738)	0.066 (0.000–1.227)	0.145 (0.100–1.242)
API without PON-1	0.304 (0.002–2.706)	0.113 (0.001–1.107)	0.248 (0.000–2.117)	0.435 (0.300–1.863)

* N = number of samples, E = epithelial tumor, S = spindle-cell mesenchymal tumor, R = round-cell mesenchymal tumor, O = other tumors.

**Table 5 vetsci-12-00533-t005:** Results at first sampling divided by neoplastic category reported as median and range (minimum–maximum).

	E *	S *	R *	*p*-Value
N * (first sampling)	10	16	27	
CRP (mg/dL)	0.895 (0.507–3.620)	0.515 (0.136–1.700)	0.779 (0.060–3.590)	0.155
Hp (mg/dL)	148.3 (43.4–173.2)	75.3 (1.0–151.2)	89.5 (1.0–173.4)	0.088
Albumin (g/L)	31.31 (19.90–35.64)	29.52 (19.36–39.40)	30.50 (16.63–37.69)	0.427
PON-1 (U/mL)	191.5 (128.5–269.2)	201.8 (104.8–281.6)	163.5 (29.0–230.3)	**0.049**
API	0.100 (0.020–1.350)	0.030 (0.000–0.390)	0.070 (0.000–1.050)	0.211
API without PON-1	0.412 (0.091–2.706)	0.15 (0.001–0.589)	0.261 (0.000–2.117)	0.106

* N = number of samples, E = epithelial tumor, S = spindle-cell mesenchymal tumor, R = round-cell mesenchymal tumor. Statistically significant *p*-values are shown in bold.

**Table 6 vetsci-12-00533-t006:** Results divided by survival time (T) reported as median and range (minimum–maximum).

	T < 30 d	T > 30 d	*p*-Value	T < 90 d	T > 90 d	*p*-Value
N *	10	45		20	35	
CRP (mg/dL)	1.530 (0.697–3.620)	0.663 (0.060–3.590)	**<0.001**	0.94 (0.194–3.620)	0.619 (0.060–3.590)	**0.011**
Hp (mg/dL)	144.1 (63.8–163.8)	77.8 (1.0–173.4)	**0.016**	135.1 (29.3–168.0)	75.4 (1.0–173.4)	0.066
Albumin (g/L)	28.34 (16.63–32.36)	30.28 (19.36–39.40)	0.771	28.59 (16.63–35.64)	30.69 (23.62–39.40)	0.284
PON-1 (U/mL)	164.3 (29.0–215.9)	183.6 (69.8–281.6)	0.156	177.9 (29.0–272.7)	181.9 (69.8–281.6)	0.687
API	0.346 (0.056–1.350)	0.054 (0.000–0.445)	**0.001**	0.114 (0.006–1.350)	0.044 (0.000–0.445)	**0.013**
API without PON-1	0.692 (0.225–2.706)	0.177 (0.000–1.673)	**<0.001**	0.395 (0.024–2.706)	0.163 (0.000–1.6732)	**0.005**

* N = number of samples. Statistically significant *p*-values are shown in bold.

**Table 7 vetsci-12-00533-t007:** AUC values derived from the ROC curves and cut-off points with the best sensitivity and specificity for each analyte.

	AUC	Cut-Off	Sensitivity	Specificity
CRP (mg/dL)	0.738	>0.977	54.55	86.84
Hp (mg/dL)	0.673	>98.1	68.18	65.79
Albumin (g/L)	0.577	≤32.36	90.91	28.95
PON-1 (U/mL)	0.611	≤132.8	31.82	94.74
API	0.716	>0.049	80.95	52.94
API without PON-1	0.731	>0.202	80.95	58.82

**Table 8 vetsci-12-00533-t008:** Absolute value and percentage of animals alive or dead depending on API and API without PON-1 cut-offs at first sampling.

	Dead		Alive	
	N	%	N	%
API > 0.049	17	51.52	16	48.48
API < 0.049	4	18.18	18	81.82
API without PON-1 > 0.202	17	54.84	14	45.16
API without PON-1 < 0.202	4	16.67	20	83.33

## Data Availability

The dataset presented in this study is available upon request from the authors.

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
