# Peer review of "Beyond Individual Acute Phase Protein Assessments: Introducing the Acute Phase Index (API) as a Prognostic Indicator in Dogs with Malignant Neoplasia"

_vetsci, 2025, doi:10.3390/vetsci12060533_

Round 1
Reviewer 1 Report
Comments and Suggestions for Authors
This is a very interesting clinical study regarding the potential importance of using the Acute Phase Index (APP) in relation with the other standard biomarkers used currently in veterinary medicine. The planning of the study is very good and the presentation of results is done in a concise and easy to intepret manner.
Some comments regarding points that could be addressed follow.
Title: Consider modifying the title by adding "in cancer patients/in dogs with malignant neoplasia etc". In its current form it does not mention for which condition API is a prognostic factor and the reader has to proceed to the abstract to find out.
Materials and methods: Since this is a prospective study, would it be better to structure it in 2 parts, a) a cross sectional study for the animals without multiple samplings and b) a longitudinal study for those monitored for some weeks?
Please add information regarding the validation of the CRP assay method for use in dogs as it is done for PON-1.
Results: As mentioned previously in the M+M section, it would be nice if there was an explanation for how many dogs were allocated in the cross sectional study and how many in the longitudinal study.
Discussion: Lines 390-393 This part can be included as a study limitation.
Table 2: Do these numbers refer to cases that might had both histology, cytology, other diagnostic tests? Please mention it.

The text quality could be improved following a professional language editing.
Author Response
Comment 1
Title: Consider modifying the title by adding "in cancer patients/in dogs with malignant neoplasia etc". In its current form it does not mention for which condition API is a prognostic factor and the reader has to proceed to the abstract to find out.
Response 1
Thank you very much for this valuable suggestion — we agree it improves clarity and relevance. We have modified the title to specify that the study was conducted in dogs with malignant neoplasia.
Comment 2
Materials and methods: Since this is a prospective study, would it be better to structure it in 2 parts, a) a cross sectional study for the animals without multiple samplings and b) a longitudinal study for those monitored for some weeks?
Please add information regarding the validation of the CRP assay method for use in dogs as it is done for PON-1.
Response 2
We appreciate your suggestion regarding the structure of this section. We have revised it accordingly, separating the description into two parts: a) a cross-sectional study for patients with a single evaluation and b) a longitudinal study for patients that were monitored over time (lines 277-282).
Additionally, we have included information regarding the validation of the CRP assay method for use in dogs, similar to what we had provided for PON-1 (lines 242-243 and 246-247).
Comment 3
Results: As mentioned previously in the M+M section, it would be nice if there was an explanation for how many dogs were allocated in the cross sectional study and how many in the longitudinal study.
Response 3
Thank you for bringing this up. As this is a prospective study, we did not define a target number of animals in advance. Dogs were included consecutively as they were presented during the study period, and this explains the final number of patients in each subgroup.
Comment 4
Discussion: Lines 390-393 This part can be included as a study limitation.
Response 4
Many thanks for this helpful remark. We have now included this point explicitly among the study limitations.
Comment 5
Table 2: Do these numbers refer to cases that might had both histology, cytology, other diagnostic tests? Please mention it.
Response 5
Thank you for the careful observation. In this table, each patient was counted only once and the numbers presented do not include cases with multiple diagnostic methods; no patients appear in more than one category.

Reviewer 2 Report
Comments and Suggestions for Authors
The scope of this manuscript was to determine the acute phase index (API) based on acute phase protein concentrations to investigate this index as a prognostic indicator in dogs with different types of neoplasia.
Comments follow:
- Title: I would suggest to include neoplasia in the title of the paper
- Simple summary: please include 'medicine' after veterinary (line 19)
- Abstract: please include 'medicine' after veterinary (line 33); please include a definition of API; I would suggest to remove the sentence "Studies linked APP variations to clinical trends in cancer" (line 31); and please consider reviewing the last sentence "These results highlight the prognostic role of API in dogs with neoplasia" as the number of animals was low when considering that there were at least 3 different tumour types and although some animals had multiple samples, these were not separated in subgroups (samples after chemotherapy?; samples from dogs without treatment? dogs underwent surgery?). I would also suggest to consider the absence of information of these multiple samples a limitation of the study (lines 427-429).
- Keywords: Please remove 'cancer' and change it to neoplasia
- Introduction: (line 49): please remove 'serum or plasma'; (lines 60-61): please consider reviewing the following sentence: "APPs are part of the innate immune response; their high sensitivity allows to use them as early markers of pathological processes (APP are markers of the inflammatory response), but their low specificity (??) ([4]) prevents them from localizing or determining the extent of a pathological process.'(it is not possible to determine the cause of the inflammatory response, however, it is possible to determine the extension of the immune response); (line 68): please change "pathological conditions" to during the acute phase response; (line 71): reference 6 is from a study with horses, so I would suggest to include studies with dogs (such as Am. J. Vet. Res., 73 (2012), pp. 34-41); (lines 93-94): please consider reviewing the following: "The literature includes only studies focusing on individual acute phase proteins (APPs). However, the alteration of a single APP does not allow to identify specific clinical situations ([19])" - as there is a number of studies that evaluated at least two or three APP, although they were not sucessful in determining prognostic or specific responses. The major suggestion for the Introduction section would be to review the content from lines 100 to 117 as there is a lot of information regarding human studies that were not useful for the discussion. Nevertheless, the information from lines 118 and 119 regarding the only studies with API in veterinary medicine were not presented/explored and are indeed useful for the discussion of results.
- Material and methods: regarding the dogs included in the study (it is in the results section "During the study period, a total of 117 samples were collected from fifty-five dogs") - should be included in line 128, and perhaps Table 3 should be Table 1 in Material and Methods order? Please include references for CRP and Haptoglobin methods validation in dogs (lines 148 and 151, respectively). Please provide a reference for 'The conversion was performed dividing PON-1 data into quartiles' (line 159) and also a reference for API calculation. Please change the word 'parameters' for analytes (in all manuscript).
- Results: although the results are relevant and with potential to further development, I would suggest to change the word indicate/indication to suggests/suggested speciaaly for the survivors and non-survivors as the number of samples was very low (10 and 12) and there are many differences between the types of neoplasia that could cause completely different acute pahse responses.
- Discussion: I would suggest to include the low number of samples/dogs as a limitation of the study as there are so may different types of neoplasia that produce so many different acute phase responses.
Author Response
Comment 1
Title: I would suggest to include neoplasia in the title of the paper
Response 1
Thank you for this helpful suggestion — we have followed your advice and revised the title accordingly to improve its clarity and focus.
Comment 2
Simple summary: please include 'medicine' after veterinary (line 19)
Response 2
Thank you for the suggestion. The term "medicine" has been added as recommended (line 20).
Comment 3
Abstract: please include 'medicine' after veterinary (line 33); please include a definition of API; I would suggest to remove the sentence "Studies linked APP variations to clinical trends in cancer" (line 31); and please consider reviewing the last sentence "These results highlight the prognostic role of API in dogs with neoplasia" as the number of animals was low when considering that there were at least 3 different tumour types and although some animals had multiple samples, these were not separated in subgroups (samples after chemotherapy?; samples from dogs without treatment? dogs underwent surgery?). I would also suggest to consider the absence of information of these multiple samples a limitation of the study (lines 427-429).
Response 3
Thank you very much for your thorough feedback. We have followed all your recommendations, including the revision of specific sentences (line 34), clarification of the API definition (lines 32-33), and a revision of the concluding sentence using more cautious language (lines 50-51)
.
Comment 4
Keywords: Please remove 'cancer' and change it to neoplasia
Response 4
Thank you. We have modified the keywords as suggested.
Comment 5
Introduction: (line 49): please remove 'serum or plasma'; (lines 60-61): please consider reviewing the following sentence: "APPs are part of the innate immune response; their high sensitivity allows to use them as early markers of pathological processes (APP are markers of the inflammatory response), but their low specificity (??) ([4]) prevents them from localizing or determining the extent of a pathological process.'(it is not possible to determine the cause of the inflammatory response, however, it is possible to determine the extension of the immune response); (line 68): please change "pathological conditions" to during the acute phase response; (line 71): reference 6 is from a study with horses, so I would suggest to include studies with dogs (such as Am. J. Vet. Res., 73 (2012), pp. 34-41); (lines 93-94): please consider reviewing the following: "The literature includes only studies focusing on individual acute phase proteins (APPs). However, the alteration of a single APP does not allow to identify specific clinical situations ([19])" - as there is a number of studies that evaluated at least two or three APP, although they were not sucessful in determining prognostic or specific responses. The major suggestion for the Introduction section would be to review the content from lines 100 to 117 as there is a lot of information regarding human studies that were not useful for the discussion. Nevertheless, the information from lines 118 and 119 regarding the only studies with API in veterinary medicine were not presented/explored and are indeed useful for the discussion of results.
Response 5
We are very grateful for your insightful and detailed recommendations. We have revised the section according to your helpful suggestions. In particular, we shortened the part focused on human literature and expanded the content related to veterinary studies to better support the discussion of our work (lines 127-151). Specifically, the details regarding the article on API in dogs with leishmaniasis had already been discussed later in the manuscript, so we moved the relevant paragraph from the discussion to introduction to avoid redundancy (lines 143-148).
Comment 6
Material and methods: regarding the dogs included in the study (it is in the results section "During the study period, a total of 117 samples were collected from fifty-five dogs") - should be included in line 128, and perhaps Table 3 should be Table 1 in Material and Methods order? Please include references for CRP and Haptoglobin methods validation in dogs (lines 148 and 151, respectively). Please provide a reference for 'The conversion was performed dividing PON-1 data into quartiles' (line 159) and also a reference for API calculation. Please change the word 'parameters' for analytes (in all manuscript).
Response 6
Thank you for these valuable comments. We clarified that, as this was a prospective study with consecutive enrollment, the number of cases was not predetermined and therefore was reported in the Results section.
The division of PON-1 values into quartiles was an arbitrary methodological choice specific to this study, so no reference is available.
We added references for API calculation (line 276) and for CRP method validation (line 247). The haptoglobin validation was conducted in-house as part of the first author's undergraduate thesis, which is cited and acknowledged in the conflict of interest statement (line 250-251).
Lastly, we have replaced "parameters" with "analytes" throughout the manuscript, as recommended.
Comment 7
Results: although the results are relevant and with potential to further development, I would suggest to change the word indicate/indication to suggests/suggested speciaaly for the survivors and non-survivors as the number of samples was very low (10 and 12) and there are many differences between the types of neoplasia that could cause completely different acute phase responses.
Response 7
Thank you for this thoughtful suggestion. We revised the language accordingly to more cautiously reflect the exploratory nature of our findings.
Comment 8
Discussion: I would suggest to include the low number of samples/dogs as a limitation of the study as there are so may different types of neoplasia that produce so many different acute phase responses.
Response 8
We fully agree, and thank you for pointing this out — the low number of cases and heterogeneity of neoplasia types are now clearly discussed as study limitations (lines 660-662).

Round 2
Reviewer 2 Report
Comments and Suggestions for Authors
Dear Authors,
Thank you for your careful response to all suggestions and the very good job in reviewing the paper. I do not have any further comments.